# OpenReview forum: "Painless Activation Steering: An Automated, Lightweight Approach for Post-Training Large Language Models"
_ICLR.cc/2026/Conference — Submitted to ICLR 2026_

### Official Review · Reviewer_p3a6 · 2025-10-31

**Soundness:** 3
**Presentation:** 2
**Contribution:** 2
**Rating:** 4
**Confidence:** 3

**Summary:**

This paper proposes a fully automated and lightweight activation steering method called PAS (Painless Activation Steering). The method leverages the model’s self-predictions on the training set to automatically construct positive and negative prompt pairs, compute in-layer activation difference vectors, and inject them during inference, thereby achieving targeted behavioral steering without modifying model weights or prompts. The authors conduct systematic experiments on three open-source models across 18 tasks. Results show that PAS yields significant and consistent improvements on behavioral tasks (e.g., bias mitigation, morality, and alignment), while providing only limited gains on intelligence-oriented tasks.

**Strengths:**

1. The method achieves automated prompt construction without human annotation, demonstrating strong scalability and compatibility with other methods. It is computationally efficient, with fast runtime, low memory cost, and high practical utility.

2. The authors performed extensive evaluation on multiple LLMs and 18 tasks, including behavioral and reasoning domains, providing a systematic view of where activation steering helps or fails, including ablation studies on steering layer, steering strength, and steering target, as well as sample-size sensitivity analyses,providing statistically solid and comprehensive results.

3. The proposed PAS can complement existing post-training methods (ICL and SFT), adding measurable improvements.

**Weaknesses:**

1. Manual tuning contradicts automation claim. The effectiveness of PAS is highly sensitive to the steering layer and steering strength, which must be determined through grid search on a validation set.

2. The authors attribute their core innovation to “achieving automation for the first time.” There have similar prior work construct steering vector automatically, such as Hyper Steer and Alpha Steer.

3. The experiment only compared three variants of PAS, lack of comparisons with other activation-guided methods.

4. The analysis remains empirical. There is no theoretical explanation or mathematical formulation clarifying why introspective PAS works or how its learned directions interact with representation space.

**Questions:**

1. The authors have demonstrated empirically that using the residual stream, medium steering strength, and middle-layer injection yields strong results. To further strengthen the paper, could the authors provide theoretical analysis or interpretive justification for these design choices, beyond empirical evidence?

2. Could an adaptive mechanism be designed to automatically select the optimal steering layer and strength, thus achieving truly full automation?

3. It is recommended to evaluate PAS under noisy data, adversarial prompts, and multilingual inputs to verify its robustness and safety in more complex scenarios.

---

> ### Author Response · Authors · 2025-11-23
> **Response to Question and Weaknessess, Part 1**
>
> > Manual tuning contradicts automation claim. The effectiveness of PAS is highly sensitive to the steering layer and steering strength, which must be determined through grid search on a validation set.
>
> We appreciate the opportunity to clarify a possible misunderstanding of our notion of **automation** and how hyperparameters are handled in PAS.
>
> **Automation refers to human effort, not the absence of hyperparameters.**
> Throughout the paper, we use “automated” to mean that, once a labeled dataset and a base LM are given, the entire pipeline—**from constructing contrastive prompts to extracting steering vectors and applying them**—is run without human prompt engineering, manual feature labeling, or calls to external frontier models. This is in contrast to prior AS methods that require **hand-crafted prompt pairs** or **manually annotated SAE features**.
>
> **Hyperparameter search is standard and fully scriptable.**
> PAS uses a simple grid search over steering layer and strength once per (model, task) pair, using a validation split. This search is itself fully automated and does not involve manual tuning. This is directly analogous to standard automated ML procedures that involve hyperparameters—for example, LASSO regression is fully automated despite requiring the selection of the tuning parameter $\lambda$ via cross-validation.
>
> In the paper, we then aggregate these experiments into **default recommendations** (middle layers, residual stream, $\lambda \approx 1$), precisely so that practitioners can use PAS *without* doing any tuning in most settings.
>
> **Empirically, PAS is not fragile to small deviations.**
> Our hyperparameter analysis (Section 3.8, Figs. 3–5, and Appendix A) shows **broad plateaus of good performance** around middle layers and moderate steering strength, rather than sharp spikes. This robustness is what allows us to provide global default settings that **transfer across tasks and models**.
>
> ---
>
> > The authors attribute their core innovation to “achieving automation for the first time.” There have similar prior work construct steering vector automatically, such as Hyper Steer and Alpha Steer.
>
> Thank you for pointing out these works. Our claim about “automation” is **task- and model-level**: PAS is designed so that, given *any* labeled dataset and *any* open-weight LM, the user can implement the entire PAS pipeline **without** manual labeling of latent features (e.g., SAE directions) or training a separate auxiliary network per behavior.
>
> By contrast:
>
> - **HyperSteer** generates a separate steering vector for each steering prompt by training a new **hypernetwork** (another transformer initialized from a truncated copy of the base LM) with a language-modeling cross-entropy loss. This requires significant additional computational budget and only produces a steering vector **tailored to a single prompt**, rather than to a general behavior pattern (e.g., sycophancy) that can only be captured by aggregating information across an entire dataset.
>
> - **AlphaSteer** is narrowly focused on conditional refusal on safety tasks, using a specialized mechanism to turn refusal steering on/off depending on input type. It assumes a **pre-existing refusal vector** (constructed via traditional hand-written prompt pairs) and does *not* propose an automated recipe to build task vectors for arbitrary benchmarks.
>
> In contrast, **PAS provides a generic, model-agnostic recipe** for running AS with **no hand-designed prompts, features, or auxiliary models**. In the revised manuscript, we will clarify our notion of “automation” to make this distinction more explicit.
>
> ---
>
> > The experiment only compared three variants of PAS, lack of comparisons with other activation-guided methods.
>
> In response to this, we have **added experiments contrasting PAS with CAA using LLM-generated prompt pairs** in the revised manuscript (Figure 3, lines 325–328). We refer the reviewer to the shared/public comment for more details.
>
> There is another family of AS methods that require **pre-labeled SAE features**. However, these methods are inherently limited in their usefulness because SAE features **do not exist for most models**. In the literature, such methods are typically evaluated only on the **Gemma** family. This makes them simply not usable for the open-weight models we study in this paper.

---

> ### Author Response · Authors · 2025-11-23
> **Response to Question and Weaknessess, Part 2**
>
> > The analysis remains empirical. There is no theoretical explanation or mathematical formulation clarifying why introspective PAS works or how its learned directions interact with representation space.
>
> While the current paper is indeed more **empirical than theoretical**, we would like to clarify what is already present and how we plan to extend it.
>
> **Existing mathematical formulation.**
> Section 2.1 introduces a formal post-training objective over evaluation dimensions and defines PAS as an intervention on internal activations via a steering vector. Section 4 discusses how injecting a fixed vector into the residual stream at layer $\ell$ is mathematically equivalent to modifying the **bias term** at that layer, placing PAS at one endpoint of a spectrum between **full weight updates** (RL/SFT) and **minimal-parameter interventions** (a single vector).
>
> **Why introspective PAS (iPAS) can help.**
> Intuitively, iPAS constructs contrastive prompts **targeted at the model’s own errors**: the positive prompts encode “how the model should behave on items it previously failed.” This is closely aligned with **error-driven, contrastive learning and self-reflection** traditions in both vision and reasoning. We already mention these connections, but we agree they could be made more explicit and will revise the text accordingly.
>
> **Limitations and future theory.**
> We explicitly state that PAS **fails to reliably improve intelligence tasks** and hypothesize that a single-layer bias shift may be too weak to induce the necessary distribution shift. As mentioned, we view this work as a **starting point** for theoretical analysis of the “Pareto frontier” between **strength of causal control** (full RL) and **parameter footprint** (a single steering vector).
>
> ---
>
> > The authors have demonstrated empirically that using the residual stream, medium steering strength, and middle-layer injection yields strong results. To further strengthen the paper, could the authors provide theoretical analysis or interpretive justification for these design choices, beyond empirical evidence?
>
> We appreciate this suggestion. The theoretical/interpretive justifications—which we have now added to the paper (lines 476–485)—are as follows:
>
> - **Residual stream.** Intervening on the residual stream is standard in the activation-steering literature, as it aggregates contributions from attention and MLP and directly feeds into the next layer. Our bias-shift view (Section 4) explicitly treats PAS as **modifying the bias in the residual pathway** at layer $\ell$, which is the most direct, architecture-agnostic location for a first-class steering primitive.
>
> - **Middle layers.** Empirically, we find that most behavior tasks are best steered by injecting in the **middle third of layers** (e.g., around layer 14 in a 32-layer LM), consistent with multiple independent observations that **high-level semantic features** (bias, sycophancy, moral framing) emerge in mid–late layers rather than at the input or output extremes. One interpretive explanation is that early layers are still processing the input and local syntax, later layers focus on surface token selection, and **conceptual reasoning about behavior and framing is represented most clearly in the middle layers**.
>
> - **Moderate strength.** Our ablations show that performance is **concave in $\lambda$**: small $\lambda$ yields small effects; very large $\lambda$ harms accuracy and can induce catastrophic forgetting; $\lambda \approx 1\text{–}5$ yields the best trade-off. This is exactly what one would expect if PAS is traversing a **semantically meaningful direction** in representation space: too little movement has little effect; too much movement overrides useful context.

---

> ### Author Response · Authors · 2025-11-23
> **Response to Question and Weaknessess, Part 3**
>
> ---
>
> > Could an adaptive mechanism be designed to automatically select the optimal steering layer and strength, thus achieving truly full automation?
>
> Our current pipeline already employs an **adaptive mechanism**: for each (model, task, method), we perform a hyperparameter grid search over (layer $\ell$, strength $\lambda$) using the validation split and choose the best-performing configuration. This tuning is **fully scripted** and requires **no manual intervention**.
>
> At the same time, to make PAS practical, we take the additional step of **learning global default hyperparameters** from these experiments—middle layers, residual stream, $\lambda \approx 1$—so that practitioners typically do *not* need to rerun this search in every new setting.
>
> ---
>
> > It is recommended to evaluate PAS under noisy data, adversarial prompts, and multilingual inputs to verify its robustness and safety in more complex scenarios.
>
> We appreciate this suggestion, but we believe these proposed settings lie **outside the scope** of PAS as framed in this work. PAS is specifically designed to **steer behavior within existing, well-curated LLM benchmarks**. Our goal is to improve model performance on these established tasks **without** retraining the model or modifying the benchmark itself.
>
> Evaluating PAS under noisy data, adversarial prompts, or multilingual inputs would target a fundamentally different research question: **model-level robustness to distribution shift or adversarial manipulation**. While these issues are important, they concern the **underlying LLM and data pipeline**, rather than the specific activation-steering mechanism we study here. We see such robustness-focused evaluations as valuable future work, but not as prerequisites for the core contribution of PAS as a lightweight, automated activation-steering tool.

---

### Official Review · Reviewer_st94 · 2025-11-01

**Soundness:** 2
**Presentation:** 3
**Contribution:** 1
**Rating:** 2
**Confidence:** 3

**Summary:**

The authors propose Painless Activation Steering (PAS), which modifies existing activation steering methods by using correct & incorrect behavior *from the model* to construct positive and negative examples, rather than using a static set of positive and negative examples.

For background, activation steering modifies the *activations* of a LM to achieve modified behavior. It involves an activation vector $a^\ast$, the steering layer $\ell$, the steering strength $\lambda$, and the target location $\operatorname{steer\textunderscore targ}$. Activation steering works by adding $\lambda\cdot a^\ast$ to the activations at location $\operatorname{steer\textunderscore targ}$ of layer $\ell$. The activation vector $a^\ast$ is constructed as the difference in the *average activation of the last token* at the location and layer of interest between two sets of examples $\mathbf{P}_k^+$ and $\mathbf{P}_k^-$, which contain examples of the desired and undesired behavior, respectively.

Now, existing work constructs $\mathbf{P}_k^+$ and $\mathbf{P}_k^-$ "manually" by combining, e.g., multiple choice questions with correct and incorrect answers. In contrast, PAS evaluates the model of interest, $M$, on the MC dataset, and uses those predictions to construct the two sets. The approaches they propose are the following:
- PAS-Full (**pasf**): $\mathbf{P}_k^+$ = examples with model-chosen correct answer, $\mathbf{P}_k^-$ = examples with model-chosen incorrect answer.
- iPAS-All (**iPASa**): Substitute the text of the answer choice in place of the answer letter.
- iPAS-Wrong-Only (**iPASwo**): Use only the set of questions the model gets wrong. $\mathbf{P}_k^+$ = examples with correct answer, $\mathbf{P}_k^-$ = examples with model-chosen incorrect answer. In this setting, the questions in $\mathbf{P}_k^+$ and $\mathbf{P}_k^-$ are the same, and only the answers are different.

The authors show that PAS is better than no steering, and achieves complementary gains over prompt-based steering (in-context learning) and weight-based steering (training).

**Strengths:**

1. Activation steering has some advantages over prompt-based and weight-based steering, which the authors discuss, making it a worthwhile direction to study for lightweight adaptation of models.
2. The authors provide extensive analysis of the different choices in activation steering, such as the steering location, layer, and other hyperparameters.

**Weaknesses:**

1. As far as I can tell, the main contribution of PAS is the choice to construct the positive and negative examples using the behavior of the model, rather than having a static set of examples of correct and incorrect behavior. The methodology itself has all been well-established in prior literature.
2. The authors do not compare PAS to the traditional method of constructing a static set of positive and negative examples. As a result, there is no evidence that PAS's approach is superior. They also do not discuss the limitation of PAS, which is that it requires constructing a different dataset for each model.
3. The authors only evaluate their method in MC settings, whereas activation steering has also been studied for open-ended generation. Constructing positive and negative examples for open-ended generation will be more difficult with their method, as it is less obvious how to divide model responses into "correct" and "incorrect" responses.
4. All the main experimental results are in the appendix, and furthermore, the results for each method (no steering, PAS, ICL+PAS, etc.) are each split into a different table! This makes the effectiveness of each approach extremely difficult to compare. As it was simply too tedious, I did not look for the empirical results supporting PAS. The only results presented in the main paper is hyperparameter analysis.

**Questions:**

1. What is the set of unconstrained evaluation dimensions $\Psi$ introduced in §2.1? This doesn't come up again in the rest of the paper.

---

> ### Author Response · Authors · 2025-11-23
> **Why We Think PAS Contains Meaningful Contribution to the Field**
>
> > As far as I can tell, the main contribution of PAS is the choice to construct the positive and negative examples using the behavior of the model, rather than having a static set of examples of correct and incorrect behavior. The methodology itself has all been well-established in prior literature.
>
> The facts stated in this critique are correct, though we argue that our paper still makes a **meaningful contribution**. Indeed, there has been work establishing that activation steering (AS) using contrastive prompt pairs works. The most well-known piece of prior literature on AS is CAA (Panickssery et al., *Steering Llama 2 via Contrastive Activation Addition*). CAA is widely regarded as an important contribution: it won an Area Chair Award at ACL 2024 and has been cited 300+ times.
>
> By the same logic given in this review, one could also argue that **CAA itself is methodologically trivial** and would not deserve the recognition it has received. After all, CAA is a follow-up to ActAdd (Turner et al., *Activation Addition: Steering Language Models Without Optimization*). One might say that CAA does not introduce fundamentally new methodology beyond what was already known from ActAdd. ActAdd also makes use of contrastive pairs of prompts (e.g., “I love talking about weddings” vs. “I hate talking about weddings,” and “I talk about weddings constantly” vs. “I do not talk about weddings constantly”; see their Table 1, Figure 1, and the paragraph starting on page 4).
>
> We do not wish to dwell on the merits or flaws of CAA here. Our point is that PAS, like CAA, does not “invent” the field of activation steering and is not a “from zero to one” step. However, in our opinion, PAS nonetheless contains **meaningful new empirical, methodological, and conceptual insights** that were not present in prior AS work, including ActAdd and CAA.
>
> **First**, we conduct, to our knowledge, the **first systematic, large-scale evaluation of AS** across 18 diverse tasks and three open-weight LMs, revealing a clear and previously unreported pattern: AS is consistently effective on **behavioral tasks** (e.g., bias, moral preference, sycophancy, alignment) but provides **no reliable benefit on intelligence/knowledge tasks**. Prior work evaluated only a small handful of behavioral settings and, as far as we know, did not establish where AS fails.
>
> We view *making this limitation explicit* as an important contribution. To the best of our knowledge, all prior work on AS (ActAdd, CAA, etc.) has focused on behavioral or stylistic settings and has not publicly reported the substantial limitations of AS on knowledge- and reasoning-oriented evaluations. Yet AS has been studied for several years. It is unlikely, in our view, that no one has ever attempted AS on OpenBookQA, ARC, LSAT, or other evaluation dimensions related to intelligence, knowledge, and reasoning; rather, the more plausible explanation is that **negative results are rarely reported because they are difficult to publish**.
>
> We believe that the community benefits when such limitations are surfaced rather than quietly circulated in small private circles. Reporting negative findings and failures can help guide researchers away from unproductive directions. Our paper thus intentionally includes the failures on intelligence-oriented tasks so that future work can build on a clearer understanding of **where AS is suitable and where it is not**.
>
> **Second**, we show that **PAS interacts with and complements existing post-training pipelines**: PAS delivers additional gains on top of ICL, and on TruthfulQA it matches or exceeds the gains of SFT while requiring orders of magnitude less computation and storage. This represents a step towards conceptualizing PAS as a **practical post-training tool for customization and personalization**, rather than merely a “cute” observational finding in mechanistic interpretability.
>
> **Finally**, PAS is, to our knowledge, **the first AS method that is fully automated and model-customized**. Driving the prompt sets directly from each model’s own success and failure patterns addresses a problem in the existing literature: a set of contrastive prompt pairs that works well for one model (the one reported in a paper) often transfers poorly to another model. This issue tends not to be visible until practitioners actually try to apply the method to their *own* models, so it is not a high-profile problem even within the mechanistic interpretability literature.
>
> PAS solves this problem by **removing the manual “prompt-tuning” step** from AS. No matter which model is used, PAS automatically constructs prompt pairs specialized to that model’s own shortcomings. PAS therefore enables **scalability across tasks, datasets, and models**, and makes AS nearly as plug-and-play as SFT or RL.
>
> Taken together, these findings help move activation steering from an **interesting proof-of-concept** to a **practical and reproducible post-training tool**.

---

> ### Author Response · Authors · 2025-11-23
>
> > The authors do not compare PAS to the traditional method of constructing a static set of positive and negative examples. As a result, there is no evidence that PAS's approach is superior.
>
> In the revised manuscript, we have **added experiments directly contrasting PAS with CAA using LLM-generated prompt pairs**. This comparison provides empirical evidence about the relative effectiveness of PAS versus a traditional static contrastive-pair approach. We refer the reviewer to the shared comment / public response for more details and the corresponding figures and tables.
>
> ---
>
> > They also do not discuss the limitation of PAS, which is that it requires constructing a different dataset for each model.
>
> All the evaluation dimensions used in the paper used the **same datasets** across different, so we assume this comment refers to the fact that different models construct **different prompt sets** from those datasets in order to obtain their steering vectors.
>
> We actually view this as an **advantage rather than a limitation**:
>
> 1. **No extra complexity.** The construction of a prompt set is *completely automated* and adds no extra computational complexity beyond standard evaluation, since the prompt set is constructed as the model processes the training set.
>
> 2. **Model-adaptive steering.** Constructing different prompt sets for different models allows PAS to **adaptively build prompt sets from each model’s own success and failure patterns**, so that the steering vector targets the idiosyncratic weaknesses of that particular model.
>
> This adaptability is precisely what makes PAS **scalable across architectures and training regimes**: unlike manually crafted contrastive pairs or SAE-based features, PAS requires **no human intervention** and **no transfer of handcrafted datasets across models**.
>
> ---
>
> > The authors only evaluate their method in MC settings, whereas activation steering has also been studied for open-ended generation. Constructing positive and negative examples for open-ended generation will be more difficult with their method, as it is less obvious how to divide model responses into "correct" and "incorrect" responses.
>
> Great point! In response, we have conducted **additional experiments on open-ended generation** using Nous-Hermes-2-Mistral and DeepSeek-R1-Distill-Llama across the 10 social-bias tasks. The new experiment and results are incorporated into the revised manuscript (**Table 12**), and full experimental details are provided in the general / shared response.
>
> These experiments explicitly demonstrate how PAS can be applied in open-ended settings, including how to assign correctness labels to responses and how the resulting steering vectors perform relative to the MCQ setting.
>
> ---
>
> > All the main experimental results are in the appendix, and furthermore, the results for each method (no steering, PAS, ICL+PAS, etc.) are each split into a different table! This makes the effectiveness of each approach extremely difficult to compare. As it was simply too tedious, I did not look for the empirical results supporting PAS. The only results presented in the main paper is hyperparameter analysis.
>
> We appreciate this concern regarding result accessibility. We follow standard empirical practice (e.g., Kalla & Broockman, 2020, *Reducing exclusionary attitudes through interpersonal conversation: Evidence from three field experiments*): **Figure 1** serves as the primary empirical summary. It presents, side by side, the causal steering effects of all three PAS variants (PASf, iPASa, iPASwo) relative to the no-steering baseline, across all **behavioral tasks**.
>
> In Section 3, we accompany each experimental setting with a brief summary and numerical support (marked in bold as **“Result”**) that captures the substantive findings for that setting. The full set of detailed estimates is placed in the appendix. Reading these summaries in the main text should be sufficient to conclude that the improvements from PAS are **large, consistent, and directly comparable at a glance**.
>
> To further improve accessibility, we will **add a consolidated results table** to the camera-ready version of the main paper, aggregating across methods, to make the comparative effectiveness of each approach even more explicit.

---

> ### Author Response · Authors · 2025-11-23
> **A Point About the Theoretical Framework**
>
> > What is the set of unconstrained evaluation dimensions introduced in §2.1? This doesn't come up again in the rest of the paper.
>
> Thank you for asking this.  We agree that the role of the **unconstrained evaluation dimensions** (denoted $\Psi$) was not sufficiently explained. Our intention in introducing $\Psi$ was to make the theoretical formulation complete and to highlight a point that is usually implicit in post-training work.
>
> In the current literature, new post-training methods are typically evaluated by showing (1) improvement on the *target* evaluation dimension and (2) no catastrophic degradation on *control* dimensions such as MMLU. However, papers rarely articulate **which dimensions the method is allowed to trade off against**. Making $\Psi$ explicit clarifies the practical reality that, for many applications, users may be willing to tolerate **decreases in certain metrics that are irrelevant to their use case**.
>
> While the implementation details and results of our paper would have made sense even without introducing $\Psi$, we included it because we anticipate that **the current era—where post-training on one domain often improves capabilities across many domains—cannot last forever**. We may be entering an era where post-training involves **inevitable trade-offs**, and being explicit about **which dimensions matter (and which do not)** will become more important as methods mature and “free lunches” disappear.
>
> In the revision (lines 205-206, 307), we clarify this motivation, make explicit how $\Psi$ fits into our setup, and tighten the connection back to the empirical sections so that readers understand why we introduce these unconstrained dimensions.
>
> Thank you for taking your time reading our manuscript and writing the review.

---

### Official Review · Reviewer_MA9J · 2025-11-02

**Soundness:** 2
**Presentation:** 3
**Contribution:** 3
**Rating:** 4
**Confidence:** 2

**Summary:**

The paper introduces a new activation steering (AS) method that automatically computes the steering factor from the training set of a target task without need of human labor. The paper shows that the method can improve open-weight model’s performance at 8B scale and can be used on top of ICL and STF on a variety of behavior tasks.

**Strengths:**

- The proposed method is cheap yet effective, and can be used jointly with other task adaption methods.
- The improvement is significant on a variety of behavior tasks, while preserving the performance on controlled tasks (MMLU).
- The paper includes ablations on the hyperparameters, including the steering strength and the position of the activation layer.

**Weaknesses:**

- Some details about the datasets used for evaluation are missing: what are the sizes of the training splits for each of the “evaluation dimensions” and if the author is calculating the activation vector using all the samples in the training splits for the main results.
- As the paper mentions, the method does not work well on intelligence-oriented tasks, which limits its application.
- The paper proposes three methods (PAsf, iPASa, iPASwo) to construct the sets of prompts to extract steering vector extraction, and the best method for different tasks is not consistent, making the question how to choose the method unexplored.
- To present the method as an alternative of ICL or SFT,  instead of showing the performance gain after applying the method on top of them, direct comparison of performance is needed.

**Questions:**

It might be more helpful to include a figure or table that demonstrates the performance gain in the main text.

---

> ### Author Response · Authors · 2025-11-23
> **Response to Question and Weaknesses, Part 1**
>
> > It might be more helpful to include a figure or table that demonstrates the performance gain in the main text.
>
> We appreciate this concern regarding result accessibility. We follow standard empirical practice (e.g., Kalla & Broockman, 2020, *Reducing exclusionary attitudes through interpersonal conversation: Evidence from three field experiments*): **Figure 1** serves as the primary empirical summary. It presents, side by side, the causal steering effects of all three PAS variants (PASf, iPASa, iPASwo) relative to the no-steering baseline, across all behavioral tasks.
>
> In Section 3, we accompany each experimental setting with a brief summary and numerical support (marked in bold as **“Result.”**) that captures the substantive findings for that setting. The full set of detailed estimates is placed in the appendix. Reading these summaries in the main text should be sufficient to conclude that the improvements from PAS are large, consistent, and directly comparable at a glance.
>
> To further improve accessibility, we will **add a consolidated results table** to the camera-ready version of the main paper to make these comparisons even more explicit.
>
> ---
>
> > Some details about the datasets used for evaluation are missing: what are the sizes of the training splits for each of the “evaluation dimensions”?
>
> The total number of available questions for each evaluation dimension differs because the underlying datasets vary in size. For the main results, we always use the ratio
>
> $$n_{\text{train}} : n_{\text{val}} : n_{\text{test}} = 3 : 1 : 1.$$
>
> In cases where a dataset is very large (e.g., there are 30,051 total questions for **Sycophancy**), we also impose a total size limit for computational convenience, since there is little additional gain in performance from using even larger data. For such datasets, we use a randomly shuffled subset of size 4,000, yielding
>
> $$n_{\text{train}} = 2400,\quad n_{\text{val}} = 800,\quad n_{\text{test}} = 800.$$
>
> In the revised manuscript, we clarify this in lines 303–304.
>
> As mentioned in Section 3.7, we have also conducted ablation experiments across a variety of data-size regimes, going down to as small as $n_{\text{train}} = 12$. We observe that **PAS remains effective in the small-data regime**. The details are in lines 422–470 and Appendix E.
>
> ---
>
> > It is unclear if the author is calculating the activation vector using all the samples in the training splits for the main results.
>
> Yes, for the main results we calculate the activation vector using **all** samples in the training split. Specifically, we take the **mean activation difference between correctly and incorrectly answered questions** in the training split.
>
> ---
>
> > As the paper mentions, the method does not work well on intelligence-oriented tasks, which limits its application.
>
> We agree that PAS is not effective on intelligence-oriented tasks. In fact, we view *making this limitation explicit* as an important contribution of the paper. To the best of our knowledge, all prior work on activation steering (e.g., ActAdd, CAA, etc.) has focused on behavioral or stylistic settings and has **not publicly reported** the substantial limitations of activation steering on knowledge- and reasoning-oriented evaluations.
>
> Activation steering has been studied for several years. It is unlikely, in our view, that no one has ever attempted AS on OpenBookQA, ARC, LSAT, or other evaluation dimensions related to intelligence, knowledge, and reasoning; rather, the more plausible explanation is that **negative results are rarely documented because they are difficult to publish**.
>
> We believe that the community benefits when such limitations are surfaced rather than quietly circulated in small private circles. Our paper therefore intentionally includes the **failures on intelligence-oriented tasks** so that future work can build on a clearer understanding of where activation steering is suitable and where it is not.
>
> Within this broader landscape, PAS is designed as a **lightweight behavioral-level post-training tool**, especially for customization and personalization scenarios where users want steerable preferences, style shifts, or alignment-related behaviors *without* weight updates. Weight-based approaches such as SFT or RL are more appropriate for improving reasoning-heavy tasks, **whereas PAS occupies a complementary niche** aimed at controllable behavioral adjustment rather than knowledge acquisition.
>
> We have revised the paper to make this motivation and viewpoint more explicit; see lines 518–525.

---

> ### Author Response · Authors · 2025-11-23
> **Response to Question and Weaknessess, Part 2**
>
> > The paper proposes three methods (PASf, iPASa, iPASwo) to construct the sets of prompts for steering vector extraction, and the best method for different tasks is not consistent, making the question of how to choose the method unexplored.
>
> As shown in **Figure 1**, the best method is **iPASwo**, which performs best on 11 out of 15 tasks. We agree that this was not highlighted clearly enough in the original submission, and we will revise the text to emphasize that **iPASwo is the most consistently strong variant** across tasks.
>
> ---
>
> > To present the method as an alternative of ICL or SFT, instead of showing the performance gain after applying the method on top of them, direct comparison of performance is needed.
>
> We did not emphasize direct comparisons because we do **not** intend PAS to be an *alternative to* ICL or SFT. That said, we *do* include direct comparisons with SFT and, in the revised version, with ICL.
>
> For **PAS versus SFT**, in the paper (lines 386–387) we write: *“Hypothesis Test C confirms that the PAS-trained model $M$ wins against the SFT-trained model $M'$ decisively…”* The result is reported in **Table 4(c)**. In short, we find **statistically significant evidence that all three PAS variants beat SFT**.
>
> For **PAS versus ICL**, in the revised manuscript (lines 366–370), we now also include direct comparisons between PAS and ICL. We find **no statistical evidence of performance differences** on social-bias or morality tasks, where the differences between PAS and ICL are small and near zero for all three methods. However, **PAS is clearly superior on alignment tasks**, with gains of roughly **+12% for iPASa, +10% for iPASwo, and +19% for PASf**, all statistically significant. **Table 11** reports full point estimates, confidence intervals, and $p$-values for PAS versus ICL.
>
> The reason we previously chose not to emphasize these direct comparisons, but instead focused on whether these methods combine well, is that we consider **ICL, SFT, and PAS not to be competitors or replacements for one another**. In contrast, if we were proposing a new variant of GRPO—say, “GRPO+”—then it would make sense to focus on comparing GRPO+ against GRPO, GRPO′, GRPO*, GRPO-2.0, and other variants, since one typically runs *one* GRPO-like algorithm at a time.
>
> In our case, however, we explicitly encourage users of PAS to **combine PAS with other post-training methods**: prompt-based (ICL), weight-based (SFT/RL), and activation-based (PAS) approaches are potentially complementary, depending on the use case. PAS can be run **on top of** ICL and SFT. Since PAS is cheap and yields additional performance gains, it is natural to combine them.
>
> We feel that the main spirit of our motivation is nicely captured in your own kind words:
>
> > *“The proposed method is cheap yet effective, and can be used jointly with other task adaption methods.”*
>
> Thank you for taking your time reading our manuscript and writing the review.

---

### Author Response · Authors · 2025-11-23
**New Experiments**

**Shared Comment**

Dear Reviewers,

We thank the reviewers for their careful and insightful reviews. We have carefully addressed all comments and revised our paper accordingly. We use blue highlighting to indicate major changes in our revised manuscript.

In the revised manuscript, we include two new experiments.

## Experiment One: PAS Beats the SOTA Baseline CAA
The first experiment (Figure 3 and lines 321–326) implements the SOTA baseline method of CAA using LLM-generated prompt pairs. We do not observe a consistent pattern, across different models, showing the effectiveness of LLM-generated CAA. While CAA leads to improvement on Sycophancy, TruthfulQA, and Common Sense, it has mixed effects across different models on the remaining 12 tasks.  This shows that PAS beats the current SOTA in AS in scalability across different models and different evaluation dimensions.

There are several reasons why the effectiveness of LLM-generated CAA is limited. First, LLM-generated content generally lacks diversity. It is well known that when asked to propose a random integer, many LLMs tend to overuse specific values (e.g., “7”), and the examples start to repeat after the first few. Second, weaker LLMs are also weaker at proposing good examples, thereby limiting the applicability of CAA. Third, when an LLM is asked to propose positive and negative examples illustrating a concept, it can only describe a subset of what it already understands about that concept; it cannot construct prompt pairs that directly target its own blind spots.

## Experiment Two: PAS Works Well On Open-Ended Generation Tasks
The second experiment (lines 394–401, Section 3.6, and Appendix F) studies the effectiveness of PAS on free-form answers to open-ended questions. For Nous-Hermes-2-Mistral and DeepSeek-R1-Distill-Llama, we reuse the 10 social-bias tasks, but strip away the multiple-choice options and present the model with a context-plus-question prompt that requires a short free-form answer.

We keep the same train/validation/test split as in the MCQ experiments. Steering vectors are still learned only from the MCQ training data using the iPASwo and iPASa methods. To emphasize that the methods are not sensitive to hyperparameters, we restrict the search space to a small grid: steering strength in $\{15, 19\}$ and intervention layer in $\{1, 4\}$. Evaluation is carried out on the open-ended test prompts. For each test instance, the model (with or without PAS steering) generates an answer via greedy decoding. We then call a separate GPT-4o grader, which receives the original context + question, the model’s answer, and the reference answer, and produces a binary correctness score (1 = correct, 0 = incorrect/insufficient). The final metric for each benchmark is the mean correctness rate across all test items, allowing a direct comparison between the unsteered model and the PAS-steered variants.

The results show that although test scores for all models drop relative to the MCQ setting—reflecting the greater challenge of open-ended generation—PAS still yields consistent performance gains across the ten tasks. Both iPAS variants outperform the unsteered models, with an average improvement of 6.2 percentage points across the two methods. Table 12 reports the full point estimates for each benchmark.

---

### Meta-Review · Area_Chair_GafZ · 2025-12-11

**Summary:**

AC believes there are multiple reviewers' concerns that have not been well addressed by the rebuttal (see below). Given the scores from reviewers in the first round of review, AC would recommend rejection.

**Reviewer Concerns:**

Reviewers' concerns that AC believes is still outstanding:

1. As the paper mentions, the method does not work well on intelligence-oriented tasks, which limits its application.

AC's comment: it seems that the authors have acknowledged this limitation.

2. The paper proposes three methods (PAsf, iPASa, iPASwo) to construct the sets of prompts to extract steering vector extraction, and the best method for different tasks is not consistent, making the question how to choose the method unexplored.

AC's comment: I think the authors fail to explain the "inconsistency" in the rebuttal.

3. Lack of direct comparison of performance with other methods.

AC's comment: This is common concerns from multiple reviewers (Reviewer MA9J and Reviewer p3a6).

4. The methodology itself has all been well-established in prior literature.

AC's comment: This is common concerns from multiple reviewers (Reviewer st94 and Reviewer p3a6).

Reviewers' concerns that AC is comfortable with:

1. The analysis remains empirical. There is no theoretical explanation or mathematical formulation clarifying why introspective PAS works or how its learned directions interact with representation space.

AC's comment: I believe theoretical explanation or mathematical formulation is not a necessary condition for a paper in ICLR.

2. Some details about the datasets used for evaluation are missing: what are the sizes of the training splits for each of the “evaluation dimensions”.

AC's comment: I think the authors have answered it in the rebuttal.

**Reviewer Scores:**

In the first round of the review, reviewers consistently vote for rejection of this paper. There are also some common concerns from multiple reviewers, such as 1) the lack of experimental comparisons, 2) the lack of discussion of the limitation of the paper, 3) similar to prior work. Therefore, AC believes the chance that reviewers change their score is very low (some reviewers even gave a score of 2) and recommends for rejection.

---

### Decision · Program_Chairs · 2026-01-26

Reject